DOI: 10.1038/s41467-018-06077-5　　**OPEN**

# Solubility-mediated sustained release enabling nitrate additive in carbonate electrolytes for stable lithium metal anode

Yayuan Liu [1], Dingchang Lin[1], Yuzhang Li[1], Guangxu Chen[1], Allen Pei [1], Oliver Nix[1], Yanbin Li[1] & Yi Cui[1,2]

The physiochemical properties of the solid-electrolyte interphase, primarily governed by electrolyte composition, have a profound impact on the electrochemical cycling of metallic lithium. Herein, we discover that the effect of nitrate anions on regulating lithium deposition previously known in ether-based electrolytes can be extended to carbonate-based systems, which dramatically alters the nuclei from dendritic to spherical, albeit extremely limited solubility. This is attributed to the preferential reduction of nitrate during solid-electrolyte interphase formation, and the mechanisms behind which are investigated based on the structure, ion-transport properties, and charge transfer kinetics of the modified interfacial environment. To overcome the solubility barrier, a solubility-mediated sustained-release methodology is introduced, in which nitrate nanoparticles are encapsulated in porous polymer gel and can be steadily dissolved during battery operation to maintain a high concentration at the electroplating front. As such, effective dendrite suppression and remarkably enhanced cycling stability are achieved in corrosive carbonate electrolytes.

[1] Department of Materials Science and Engineering, Stanford University, Stanford, CA 94305, USA. [2] Stanford Institute for Materials and Energy Sciences, SLAC National Accelerator Laboratory, 2575 Sand Hill Road, Menlo Park, CA 94025, USA. Correspondence and requests for materials should be addressed to Y.C. (email: yicui@stanford.edu)

The morphology of electrodeposited metals is dictated to a great extent by the physiochemical characteristics at the metal/electrolyte interface, including ion transport, interfacial energy, and mechanics, etc.[1]. This especially stands for the electrochemical plating of metallic lithium (Li), the ultimate anode for next-generation batteries with highest specific capacity (3860 mAh g$^{-1}$) and lowest electrode potential of all possible alternatives[2,3]. In particular, the electrodeposition of Li is complicated by the instantaneous formation of a resistive interfacial passivation (i.e. the solid-electrolyte interphase, SEI), originated from the parasitic reduction of electrolyte components by the highly reactive Li[4]. The chemical heterogeneity and mechanical instability of the SEI layer is generally believed to induce non-uniform ion flux, resulting in the formation of Li dendrites that could lead to internal short circuit and compromise battery safety, while the repeated breakdown and repair of SEI brings about continuous loss of active materials, giving rise to limited battery cycle life[5].

Extensive research has been devoted to regulate the surface reactivity of Li metal. Among all the tactics explored to date (protective coatings[6,7], nanostructured electrodes[8,9], high-modulus separators[10,11], etc.), tailoring the electrolyte composition is among the most essential and prominent paradigms, for it can directly impact the physiochemistries of the SEI layer, modifying the interfacial environment to alter Li deposition behavior[12–16].

It is known that the choice of electrolyte solvents can already offer pronounced effect on Li deposition uniformity. In general, ether-based electrolytes demonstrate relatively controlled deposition and thus high Coulombic efficiency (CE), ascribed to the formation of oligomeric SEI with superior flexibility[17]. The application of ether electrolytes, however, is severely hampered by its high flammability and low oxidation potential (anodic decomposition at <4 V vs Li$^+$/Li)[18], mismatched with the safety benchmark and the emerging high-voltage cathodes[19]. On the other hand, carbonate-based electrolytes exhibit lower flammability and higher anodic potential, and, therefore, are used exclusively in almost all the commercial Li-ion batteries. Nevertheless, they are highly corrosive to metallic Li, rendering aggravated dendrite formation and poor cycling efficiency. Fluorinated molecules, such as fluoroethylene carbonate (FEC), are frequently adopted as film-forming additives in carbonates, which is believed to promote the formation of LiF on Li surface as a favorable SEI passivation component[20–22]. Even with these additives, the Li anode CE remains deficient for practical applications, and the adverse effects of FEC decomposition at elevated temperatures bring concerns when being used in full-cells[23]. High salt concentration has recently arisen as another exemplary method to stabilize Li metal in carbonate electrolytes, the effect of which is originated from the unique solvation structure that reduces the solvent reactivity[24–26]. Nevertheless, its economic effectiveness, rheological properties, and ion transport need to be evaluated for practical applications. Thus, it is apparent that the development of new electrolyte reformulation strategies, which could circumvent the potential drawbacks of the existing ones, remains highly desirable.

Theoretically, if an additive is much more liable towards reduction than other electrolyte components to form an interfacial environment favorable for Li deposition, good electrochemical performance can be achieved regardless of the electrolyte systems ("electrolyte-agnostic" additive). Herein, we report the discovery of the pronounced effect of nitrate anions (NO$_3^-$) on the morphology and reversibility of Li deposition in carbonate-based electrolytes. Although nitrate additive is not unfamiliar to the community, as LiNO$_3$ is regularly used in ethers for Li-sulfur batteries[27,28], it remains being unexplored in carbonates partially due to its extremely low solubility. However, we observe that the preferential reduction of NO$_3^-$ during SEI formation in carbonates, albeit at very low concentration, could substantially alter the interfacial chemistry, resulting in spherical Li nuclei (instead of dendritic) and greatly improved CE exceeding the values in ethers. To further overcome the solubility limitation, a solubility-mediated sustained-release concept is proposed, where LiNO$_3$ nanoparticles are dispersed in porous polymer backbone on anode surface and can be steadily dissolved during battery operation when soluble LiNO$_3$ is consumed. With the maintained concentration of NO$_3^-$ on Li surface frontier, the anode CE can exceed 98% for over 200 cycles, and the cycle life of full-cells paired with LiNi$_{1/3}$Mn$_{1/3}$Co$_{1/3}$O$_2$ (NMC) is more than quadrupled, which is appreciable in corrosive carbonate electrolytes. Note that during the peer-review process of our manuscript, we noticed another work utilizing a slightly similar engineering method to overcome the solubility barrier of LiNO$_3$ in carbonate electrolyte, which also resulted in improved Li anode CE[29]. Nevertheless, we are confident that our comprehensive work brings a significant amount of new knowledge complementary to the other study. Particularly, the mechanism of NO$_3^-$ in modifying the SEI properties is studied deliberately on the basis of cryo-electron microscopy (cryo-EM)[30,31], ultramicroelectrode, X-ray photoelectron spectroscopy (XPS), and electrochemical impedance spectroscopy (EIS). The characterizations unravel a distinct structural change of the SEI from amorphous to bilayered configuration, a strong presence of nitrogen-containing species inside the SEI, and a greatly increased exchange current density of the Li$^+$/Li couple with the addition of NO$_3^-$, and the effects of which are discussed in detail. This work not only enables uniform Li deposition with high reversibility in typical carbonate-based electrolytes without compromising the stability, ionic conductivity, viscosity, and cost of the electrolytes, but also provides fruitful insights on the exact role of nitrate additive at a fundamental level.

## Results

**Effects of nitrate on Li morphology in carbonate electrolyte.** As frequently mentioned in the literature, the cathodic decomposition of NO$_3^-$ can start as early as ~1.7 V vs Li$^+$/Li[32], which is much higher than other carbonate-based electrolyte components, making it a promising candidate as "electrolyte-agnostic" additive (Fig. 1a)[33]. Therefore, if the preferential reduction of NO$_3^-$ can yield an SEI conducive to controlled Li deposition, the presence of NO$_3^-$ in carbonate electrolytes is highly likely to afford improved Li metal performance. To determine the exact reduction potential of NO$_3^-$, cyclic voltammogram (CV) of 1.0 M LiNO$_3$ was obtained on a stainless-steel electrode at a scanning rate of 0.1 mV s$^{-1}$. Dimethoxyethane (DME) was selected as the solvent for the CV scan, due to its stability against reduction (–1.68 V vs Li$^+$/Li) and high LiNO$_3$ solubility[34]. As can be observed in Fig. 1b, there is a deep cathodic current peak beginning at ~1.69 V vs Li/Li$^+$ in the first CV cycle, which disappeared upon the second scan. And no similar reduction peak can be observed in the first CV cycle when LiNO$_3$ was replaced by lithium bis(trifluoromethanesulfonyl)imide (LiTFSI). Thus, the irreversible cathodic current starting at ~1.69 V vs Li/Li$^+$ can be assigned with confidence to the reduction of NO$_3^-$. The easy reduction of NO$_3^-$ participates as the frontier reaction during SEI formation and plays a decisive role in the physiochemical properties of the SEI. For this reason, the addition of LiNO$_3$ in ether electrolytes can drastically change the Li deposition morphology from irregular to densely packed spheres, together with much improved CE (Supplementary Fig. 1).

To explore the effect of NO$_3^-$ in carbonate-based electrolytes, Li deposition on copper (Cu) foil was carried out using conventional 1.0 M lithium hexafluorophosphate (LiPF$_6$) in 1:1

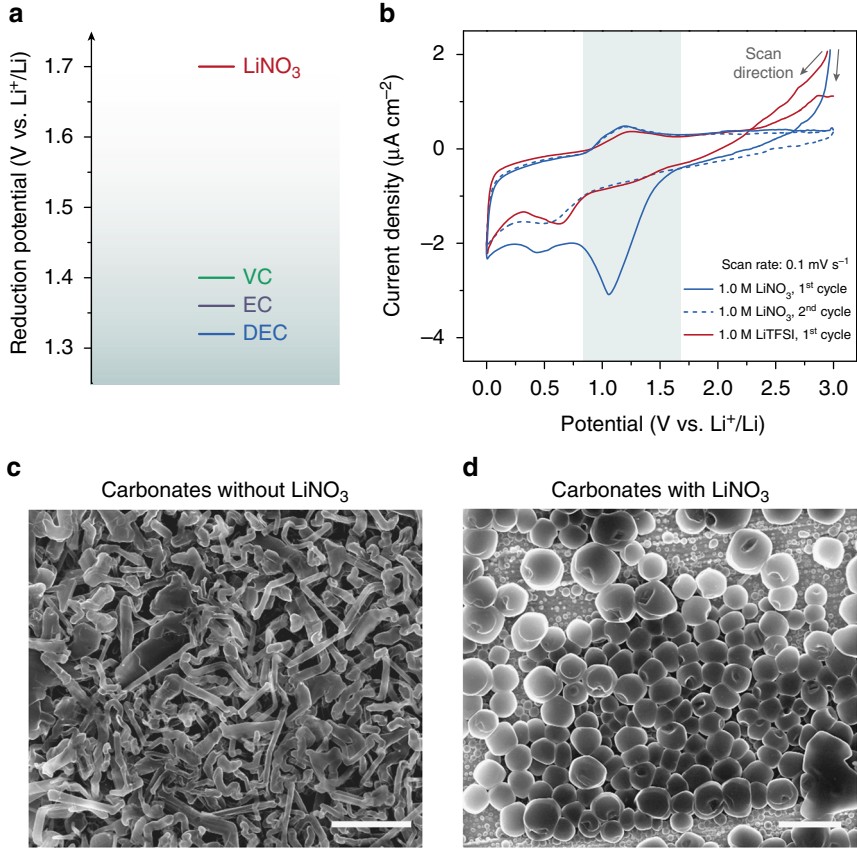

**Fig. 1** Preferential reduction of nitrate in carbonate electrolytes alters the Li nucleation morphology. **a** Experimental reduction potential of various electrolyte components (common solvents like EC and DEC; typical additive VC) on inert electrode reported in the literature[31]. **b** CV of DME-based electrolytes with 1.0 M LiNO₃ or 1.0 M LiTFSI at a scan rate of 0.1 mV s⁻¹. Stainless-steel was used as the working electrode and Li foil was used as the counter/reference electrode. SEM images of the Li nucleation morphology in 1.0 M LiPF₆ EC/DEC electrolyte **c** without and **d** with saturated LiNO₃. Scale bars, 2 µm. The deposition was carried out at a current density of 1 mA cm⁻² and a capacity of 0.1 mAh cm⁻²

v/v ethylene carbonate (EC)/diethyl carbonate (DEC) electrolyte saturated with LiNO₃. To our great surprise, although dendritic Li was observed in neat carbonate electrolyte at as early as the nucleation stage (0.1 mAh cm⁻²; Fig. 1c), uniform Li particles can be obtained in the presence of LiNO₃ (Fig. 1d). Such impact on morphology is far more appreciable than conventional film-forming additives such as FEC and vinylene carbonate (VC), with which the Li deposition remained wire-shaped (Supplementary Fig. 2). Noticeably, the effect is indifference to the selection of cations and can be extended to other molecular species with N–O bond (e.g. tetradodecylammonium nitrate and sodium nitrite; Supplementary Fig. 3). The additive is also generally applicable to different carbonate systems (e.g. LiTFSI in propylene carbonate, PC, Supplementary Fig. 4).

In spite of the pronounced effect of NO₃⁻ on Li nucleation morphology, the practicality of the additive is severely limited by its extremely low solubility in carbonates. Since NO₃⁻ needs to be continuously consumed during SEI formation, its control over Li plating wears off quickly with increased deposition capacity (Supplementary Fig. 4).

**Overcoming the solubility barrier via sustained release.** The maximum concentration of NO₃⁻ in various carbonate-based electrolytes was determined via colorimetry by cadmium reduction using a discrete analyzer (Fig. 2a). The solubility of LiNO₃ in EC/DEC-based electrolytes is ~800 ppm (equivalent to ~0.012 M), which is at least one order of magnitude smaller than the

additive concentration in ether electrolyte[27]. Reducing the LiPF₆ concentration would increase the NO₃⁻ solubility, ascribed to the "common-ion effect". Though the dielectric constants of cyclic carbonate solvents are much higher than ethers, which intuitively shall result in better salt-dissolving capability, salt dissociation is also highly dependent upon the Gutmann donor number (DN) of the solvent, a parameter measuring the Lewis basicity[35]. In brief, for salt dissociation to occur, the electron donation from the solvent molecule to the cation has to exceed the electronic interaction between the cation and anion itself, which otherwise leads to strong contact ions pairs (Supplementary Fig. 5). However, the DN of cyclic carbonates (~15) is much lower than that of NO₃⁻ (~22) and DME (~20)[36,37], accounting for the low solubility of nitrate salts in carbonates.

To overcome the solubility limitation and render nitrate additive applicable in carbonates, we propose the strategy of solubility-mediated sustained release (Fig. 2b). Specifically, a freestanding membrane consists of LiNO₃ dissolved in a polymeric matrix (termed LiNO₃ sustained-release film, LNO-SRF) was introduced on top of the anode, such that NO₃⁻ can be continuously replenished from the electrolyte-swelled polymer gel during Li deposition to maintain an appreciable local NO₃⁻ concentration at the anode. Freestanding LNO-SRF was fabricated by blade-coating a mixture of LiNO₃ and poly(vinylidene fluoride-co-hexafluoropropylene) (PVDF-HFP) in acetone on a rough metal surface, followed by peeling-off. It could also be coated on battery separators for large-scale applications, which is compatible to the state-of-art battery manufacturing process.

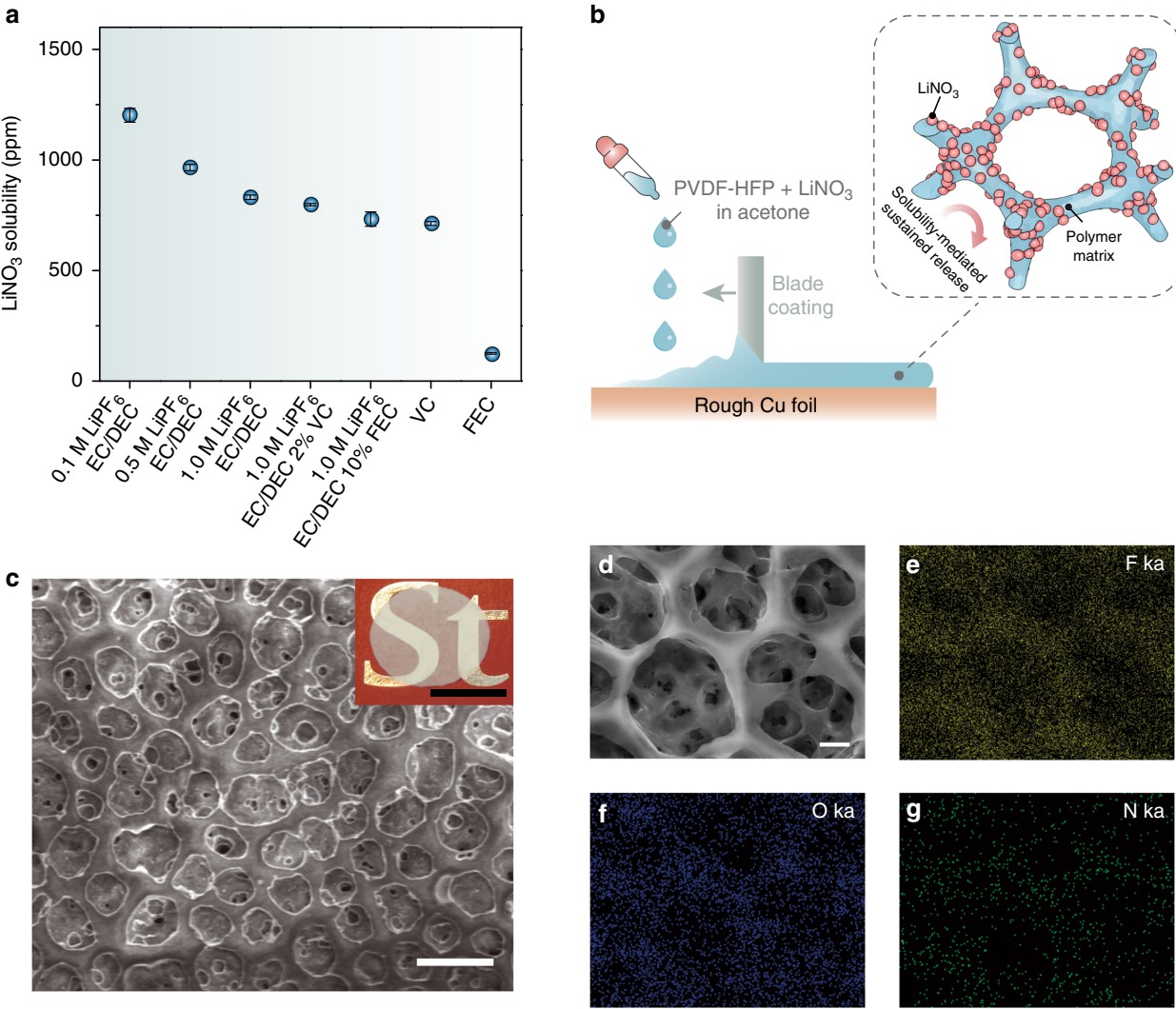

**Fig. 2** Fabrication of the solubility-mediated sustained-release film. **a** Solubility of LiNO$_3$ in EC/DEC-based electrolytes with different LiPF$_6$ concentrations and additives, in pure VC and in pure FEC. **b** Schematic illustrating the fabrication and working principle of LNO-SRF. **c** Top-view SEM image and photo image (inset) of LNO-SRF. Scale bar, 5 μm and inset scale bar, 1 cm. **d** High-magnification SEM image and **e**–**g** the corresponding EDX elemental mappings of LNO-SRF. Scale bar, 2 μm

Figure 2c shows the typical scanning electron microscopy (SEM) image of LNO-SRF, revealing a highly porous structure likely caused by the phase inversion of PVDF-HFP during fabrication in the presence of hygroscopic LiNO$_3$[38]. No discernable salt crystals/agglomerates were observed from high-magnification SEM image (Fig. 2d, Supplementary Fig. 6 shows the typical morphology of LiNO$_3$ crystals), and uniform nitrogen distribution can be confirmed by energy-dispersive X-ray (EDX) elemental mapping (Fig. 2e–g). The high film porosity and the fine dispersion of LiNO$_3$ species are particularly desirable for electrolyte wetting and rapid NO$_3^-$ release during battery operation. Thus, our method is far superior to directly adding LiNO$_3$ solid into the electrolyte, which has been tested inefficacious for improving Li metal performance, due to inhomogeneous salt distribution (sedimentation) and slow dissolution kinetics of the large particles (Supplementary Fig. 7). The typical thickness of the membrane is ~18 μm, although much thinner film can also be prepared in practical batteries, and distinct LiNO$_3$ peaks were observed from the X-ray diffraction (XRD) pattern, confirming the high mass loading of LiNO$_3$ inside the LNO-SRF (Supplementary Fig. 8 and Supplementary Method 1; the particle size was calculated to be ~60 nm from the

XRD data according to the Scherrer equation). The obtained LNO-SRF was dried inside an argon-filled glovebox with sub-ppm water and oxygen level at 120 °C for at least 1 week before use, in order to prevent the introduction of water contamination.

**High efficiency cycling in half- and full-cell configurations**. Figure 3a–c shows the morphological evolution of Li with increasing deposition capacities in the presence of LNO-SRF. Instead of growing into dendrites, as in the case of carbonates saturated with NO$_3^-$, the densely packed spherical Li nuclei evolved into nodule-like shapes with increasing capacity (Supplementary Fig.9). Moreover, the modulation effect can be preserved over repeated plating/stripping cycles on both Li foil (Fig. 3d) and Cu substrate (Supplementary Fig.10). Since the observation cannot be made using neat PVDF-HFP membrane alone (Supplementary Fig.11), such sustained control over Li deposition morphology shall be attributed to the fast, steady release of NO$_3^-$ from the microporous LNO-SRF, enabling a reasonable local additive concentration at the electroplating front despite its low solubility (Fig. 3e).

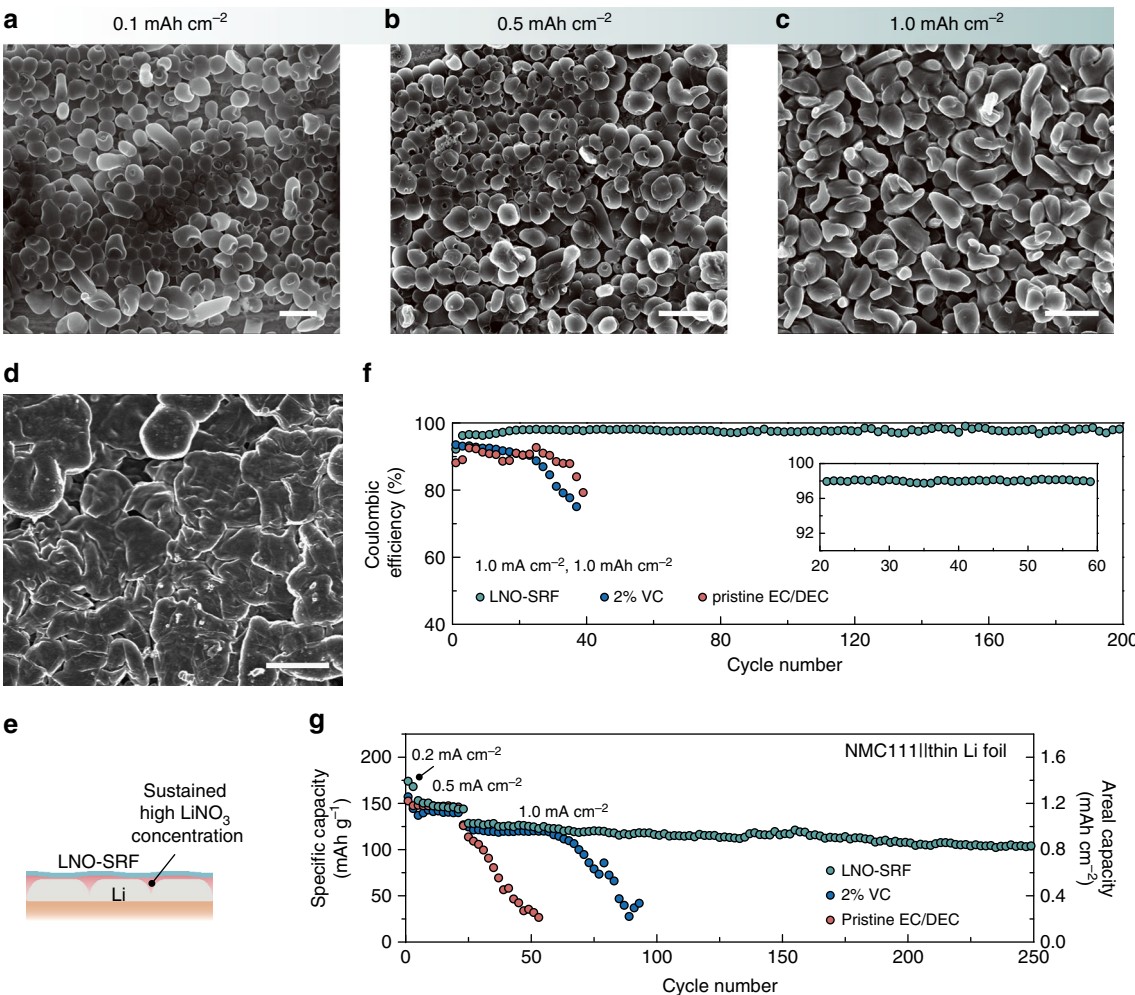

**Fig. 3** Deposition morphology and cycling CE with LNO-SRF. SEM images of the Li deposition morphology on Cu substrate covered by LNO-SRF at a current density of 1 mA cm⁻² and a capacity of **a** 0.1 mAh cm⁻², **b** 0.5 mAh cm⁻², and **c** 1 mAh cm⁻², respectively. **d** SEM image of the Li foil surface after 50 stripping/plating cycles in a Li||Li symmetric cell with LNO-SRF. The cycling was done at a current density of 1 mA cm⁻² and a capacity of 1 mAh cm⁻². Scale bars, 2 μm in **a** and 5 μm in **b**–**d**. **e** The solubility-mediated dissolution of LiNO₃ from the microporous LNO-SRF can maintain a high local NO₃⁻ concentration at the anode surface during battery cycling, which therefore results in a sustained modulation of Li deposition morphology. **f** Li cycling CE at a current density of 1 mA cm⁻² and a capacity of 1 mAh cm⁻² on Cu without additives (pristine EC/DEC), and with VC (2% VC) or nitrate (LNO-SRF) as additive. The electrolyte used in all cases were 0.5 M LiPF₆ EC/DEC. **g** Long-term cycling performance of Li||NMC full-cell with a limited amount of excess Li (42 μm thin Li foil) in 0.5 M LiPF₆ EC/DEC electrolyte (pristine EC/DEC) and in 0.5 M LiPF₆ EC/DEC electrolyte with VC (2% VC) or nitrate (LNO-SRF) as additive. The areal mass loading of NMC was ~8 mg cm⁻². The first two formation cycles were carried out at a current density of 0.2 mA cm⁻², followed by 20 cycles at 0.5 mA cm⁻², and the long-term cycling was at 1.0 mA cm⁻²

The Li anode cycling efficiency was determined by galvanostatic Li plating/stripping on Cu electrode at various current densities and capacities. Since reducing the LiPF₆ concentration results in higher NO₃⁻ solubility, 0.5 M LiPF₆ in EC/DEC was chosen as the electrolyte. Further reducing the LiPF₆ concentration was found to undermine the Li cycling efficiency (Supplementary Fig. 12). At a current density of 0.5 mA cm⁻² and a capacity of 0.5 mAh cm⁻², the CE on Cu electrode covered with LNO-SRF was indeed higher in 0.5 M LiPF₆ EC/DEC electrolyte than in the 1.0 M counterpart (Supplementary Fig. 13), thanks to the increased additive concentration. Nevertheless, the reversibility of both cells was significantly better than those with bare Cu electrode or electrode covered with pure PVDF-HFP membrane, along with a much improved first-cycle CE (from ~85% on bare Cu to >92%; Supplementary Fig. 13).

Importantly, nitrate additive enabled by LNO-SRF considerably outperformed the conventional additives over extended cycling, promoting the Li cycling efficiency and stability in carbonate electrolytes to a level comparable to, if not better than, ether electrolytes (Fig. 3f and Supplementary Fig. 14). At a current density of 1 mA cm⁻² and a capacity of 1 mAh cm⁻², the LNO-SRF cell demonstrated high efficiency cycling stable over 200 cycles (averaged ~98.1% exclusive of the activation cycles), while the Li CE in EC/DEC electrolyte with 2 wt% VC started at ~93% and decayed rapidly within 40 cycles. Since additives such as FEC and VC fail to improve the deposition morphology, high-surface-area Li dendrites can lead to non-negligible side reactions and excess "dead lithium" formation in each plating/stripping cycle[39], which account for the fast CE fading. On the other hand, nitrate additive results in nodule-like Li deposits with reduced surface area (and more stable SEI as will be discussed later), which is beneficial for reversible Li cycling. This can be further confirmed from the stable Li deposition polarization during prolonged cycling (Supplementary Fig. 15). Moreover, the solubility-mediated sustained release of NO₃⁻ is generally applicable to carbonate-based electrolytes (Supplementary Fig. 16), and compromises neither the anodic stability nor the

impedance of the cell (Supplementary Fig. 17), further corroborating the advantages of LNO-SRF.

To demonstrate that LNO-SRF in carbonate electrolytes can lead to more competitive Li metal batteries, we constructed full-cells with NMC as the cathode. Notably, only a finite amount of Li reservoir was employed in this study (either 42 μm Li foil or 150 μm three-dimensional Li metal-reduced graphene oxide composite anode, Li-rGO; Supplementary Fig. 18)[8]. Assuming the cathode has decent reversibility, the capacity fading of the full-cell shall be associated mainly to the loss of Li at the anode side (Supplementary Fig. 19). At a cathode mass loading of 8 mg cm$^{-2}$, the cell cycled in pristine 0.5 M LiPF$_6$ EC/DEC electrolyte exhibited an extremely limited reversibility, the capacity of which started to decay rapidly after merely 25 cycles due to the depletion of Li metal (Fig. 3g). The addition of 2% VC moderately improved the electrochemical performance of the full-cell, extending stable cycling to ~60 cycles. However, the life time of the Li‖NMC cells can be more than quadrupled with the introduction of LNO-SRF, which demonstrated little capacity decay up to >250 cycles. Moreover, when the NMC cathode with commercial-level mass loading (~20 mg cm$^{-2}$) was paired with three-dimensional Li anode (Li-rGO composite)[8], the electrochemical cycling stability of the full-cell with LNO-SRF can even outperform the one with VC additive and nearly-infinite Li reservoir (Supplementary Fig. 20), further confirming the unique advantages of LNO-SRF over conventional additives in carbonate electrolytes. Admittedly, the reported cycling stability still falls short of the stringent requirements of commercial Li-ion batteries. Nevertheless, the effectiveness and practicality of our proposed solubility-mediated sustained nitrate release strategy in improving the safe and efficient operation of the Li metal anode is clear from the electrochemical results. We are positive that future improvements can be expected when LNO-SRF is used in conjunction with other strategies that have been proven efficacious for Li metal stabilization, such as nanostructuring the electrode[40], and special electrochemical cycling protocols[41].

## Discussion

To elucidate the mechanism behind the improved morphology and electrochemical performance of Li metal anode with nitrate additive, which is likely originated from the modified SEI properties, we first resolved the structure of the SEI via transmission electron microscopy (TEM) at cryogenic temperature[30]. The Cryo-EM technique was recently developed in our lab to successfully visualize fragile battery materials with atomic scale resolution at their pristine states without damage. Li was deposited on Cu TEM grid using standard battery condition (Supplementary Fig. 21) and loaded into TEM strictly following the cryo-transfer protocol without air exposure, such that the structural and chemical information of the SEI can be preserved at its native state. Figure 4a is a typical cryo-EM image of Li nuclei deposited in carbonate electrolyte with NO$_3^-$, and the edge of the particles with darker contrast corresponds to the SEI region (due to the higher atomic numbers of the SEI components than elemental Li). Lattice spacings matching the {110} planes of metallic Li can be clearly resolved (Fig. 4b), confirming the crystallinity of the electroplated Li. Importantly, high-resolution cryo-EM images of the SEI clearly revealed a bilayered structure (Fig. 4c, d and Supplementary Fig. 22). The outer layer consisted of highly ordered crystalline Li$_2$O, while the inner layer appeared as an amorphous matrix with nanocrystallites dispersed inside (mainly Li$_2$O and Li$_2$CO$_3$). Energy-filtered TEM images (Fig. 4e) confirmed the high concentration of Li at the inside of the deposits (metallic Li), the even distribution of carbon throughout the SEI (Li$_2$CO$_3$, alkyl carbonates, polymeric compounds from solvent

decomposition, etc.), and the predominant localization of oxygen on the outside due to the dense Li$_2$O shell. Although no crystalline phases containing nitrogen can be observed, nitrogen can indeed be detected over the SEI region via the electron energy loss spectroscopy (EELS, Supplementary Fig. 23). Thus, the nitrogen species from the preferential reduction of NO$_3^-$ shall remain amorphous in the SEI (or the size of their domains are too small to be observed by TEM possibly due to limited concentration). This bilayered SEI configuration stands a stark contrast to the mosaic model of SEI in pristine EC/DEC electrolyte reported previously[30], where only the amorphous matrix with dispersed nanocrystals was seen (Fig. 4f, g).

One hypothesis to account for the formation of the distinct bilayered structure is that, during the SEI formation, NO$_3^-$ is reduced first to various insoluble N$_x$O$_y^-$ species and Li$_2$O[42,43]. The crystallization of Li$_2$O is kinetically favorable, forming a dense crystalline layer[44], while the N$_x$O$_y^-$ species remain largely amorphous. However, this initial SEI layer cannot completely prevent the permeation of electrolyte given the volume change of Li metal during deposition, such that additional reactions between the solvents and Li result in the amorphous inner layer, until the electron tunneling limit is reached. This observation is in line with previous studies based on surface-sensitive techniques[45].

The dense, uniform inorganic SEI outer layer can profoundly improve the homogeneity of the SEI, which not only promotes even Li-ion flux during deposition to suppress dendrites, but also effectively shields the Li surface from electrolyte corrosion to improve the cycling efficiency. Note that although FEC additive also results in a bilayered SEI structure as reported previously[30], it does not form spherical Li nuclei. Therefore, the exact Li deposition morphology also relies heavily on the physiochemical properties (ion-transport kinetics, interfacial energy, etc.) of the amorphous components of the SEI, which is beyond the information that can be provided from cryo-EM.

To correlate the formation of spherical Li with the chemical compositions of the SEI, XPS analysis with step-by-step sputtering was carried out on electrodeposited Li nuclei in EC/DEC electrolyte with nitrate additive. Consistent with the EELS data, XPS showed clear N$_{1s}$ signal in the SEI (Fig. 5a). The surface nitrogen species possessed higher oxidation states (LiN$_x$O$_y$), while more Li$_3$N can be detected closer to the Li side. Compare to the SEI formed in neat EC/DEC electrolyte, no other significant compositional differences can be detected in XPS, apart from slight concentration variations of different SEI components and a higher LiF content (Supplementary Figs. 24–27). Therefore, the nitrogen species from the preferential reduction of NO$_3^-$ can naturally be interpreted to have a profound effect on the Li surface chemistry, likely by changing the ion-transport property of the SEI, since Li$_3$N has long been known to exhibit superior Li-ion conductivity[46]. Such speculation can be verified via EIS measurement on Li‖Li symmetric cells (Supplementary Fig. 28). It was found that FEC and VC resulted in a more resistive SEI, possibly due to the high LiF content with limited Li-ion conductivity, and the cell impedance increased gradually with time[47]. However, the presence of NO$_3^-$ substantially reduced the impedance of Li electrodes and the values slightly decreased upon storage, which can be ascribed to the further reduction of LiN$_x$O$_y$ to Li$_3$N. The low-resistance SEI layer would increase the probability of Li adatom insertion on the bulk surface, suppressing deposition on the tips/kinks of existing protrusions to obstruct dendrite formation[48], which account for the unique spherical Li nuclei observed in the presence of nitrate additive in carbonate electrolytes.

Finally, the exchange current of the Li$^+$/Li couple in carbonate electrolyte with nitrate additive was measured using ultra-microelectrode, in an effort to shed light on the improved electrochemical performance from the charge transfer kinetics

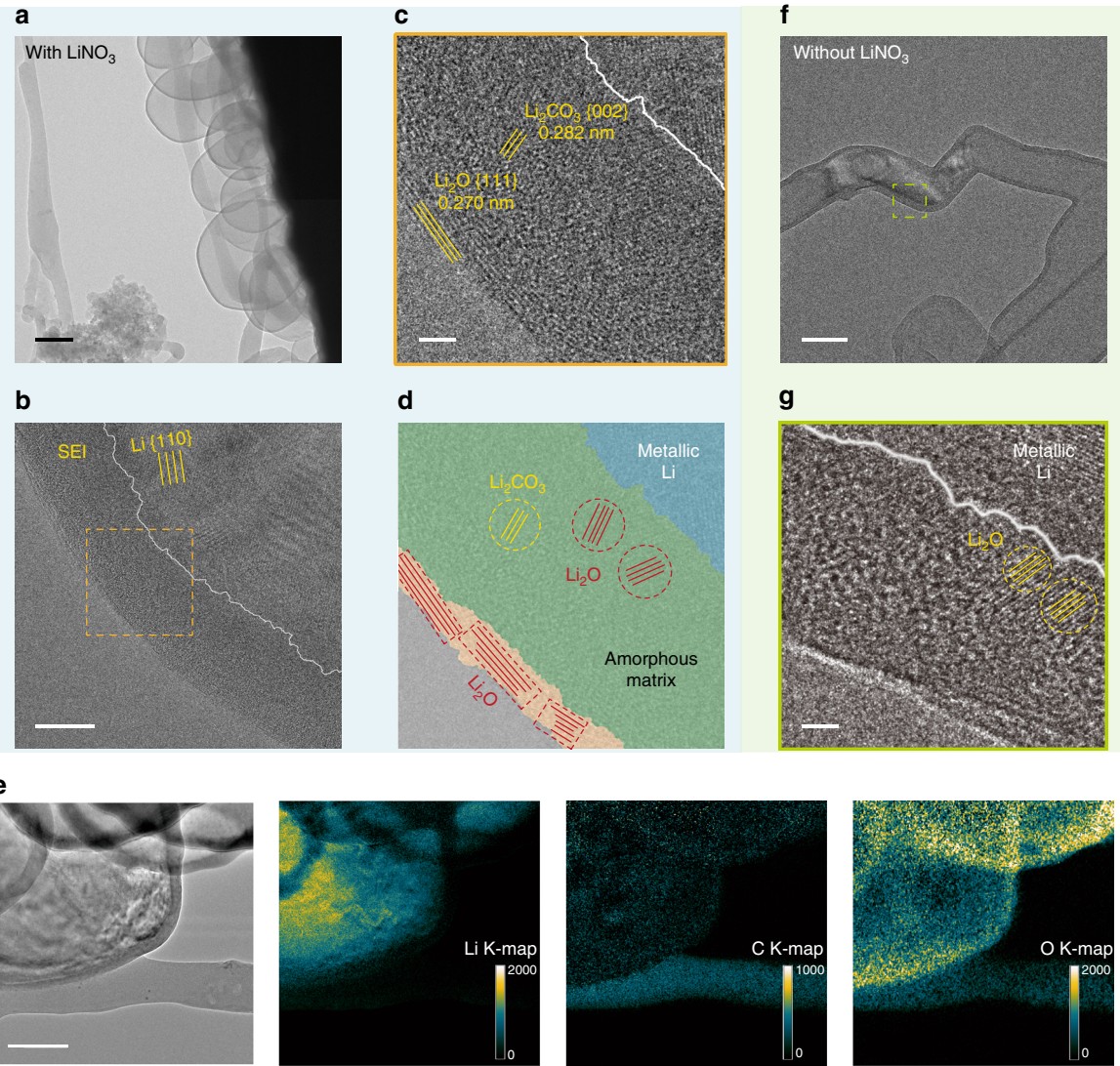

**Fig. 4** Cryo-EM study on the SEI structure in the presence of nitrate. **a** Low-magnification and **b** high-resolution TEM images of Li deposits in 0.5 M LiPF$_6$ EC/DEC electrolyte saturated with NO$_3^-$ at a current density of 0.5 mA cm$^{-2}$ for 150 s. The white line in **b** delineates the boundary between metallic Li and SEI. Scale bars, 200 nm in **a** and 10 nm in **b**. **c** Magnified TEM image and **d** the corresponding schematic of the orange region outlined in **b** showing the bilayered structure of SEI. The outside is a dense layer of Li$_2$O displaying clear lattice fringes, and the inside is an amorphous matrix with crystalline domains (Li$_2$O and Li$_2$CO$_3$) dispersed randomly throughout. Scale bar, 2 nm in **c**. **e** Energy-filtered TEM mapping of elemental distributions of Li, C, and O across the Li deposits and the corresponding unfiltered TEM image of the region. Scale bar, 100 nm. **f** Low-magnification and **g** high-resolution TEM images of Li deposits in 0.5 M LiPF$_6$ EC/DEC electrolyte without NO$_3^-$. The SEI showed a mosaic model with crystalline domains dispersed in an amorphous matrix. Scale bars, 100 nm in **f** and 2 nm in **g**

perspective (Supplementary Method 2). Ultramicroelectrode (tungsten wire 25 μm in diameter) was employed such that Li can be freshly deposited and then immediately stripped at a rate fast enough to minimize the interference from the Li surface films, and high-quality data can be obtained without being distorted by the solution ohmic drop[49]. As calculated from the CV scans shown in Fig. 5b, the exchange current of Li$^+$/Li in 1.0 M LiPF$_6$ EC/DEC electrolyte was on average 33.5 mA cm$^{-2}$, while the value increased by more than 50% when the electrolyte is saturated with NO$_3^-$ (52.1 mA cm$^{-2}$). Moreover, with the addition of nitrate additive, a much lower (more than 100 mV) nucleation onset potential can be observed from the CV scans, further indicates the improved charge transfer kinetics (Fig. 5c).

The increased exchange current reinforces our conclusions from the EIS results and is in line with the observation of a more controlled Li deposition morphology with nitrate additive, since higher exchange current tends to result in larger-sized Li deposits. Specifically, metal electrodeposition on inert electrodes begins with the formation of separate growth centers, and once a nucleus has been formed, the passing current causes a local deformation of the electric field, resulting in a potential drop in its vicinity that prevent the occurrence of new nucleation events (i.e. the nucleation exclusion zone). High exchange current serves to increase the size of the nucleation exclusion zones, which can reduce the nucleation rate to form bigger Li particles with reduced surface area[50]. To mechanistically explain the enhanced exchange current measured by ultramicroelectrode, we speculate that, due to the high DN, nitrate anions can partially replace the solvent molecules in the Li$^+$ solvation structure, and therefore substantially weakens the interaction strength between Li$^+$ and solvents to facilitate ion desolvation, which is considered as the rate-limiting step in the charge transfer process[51].

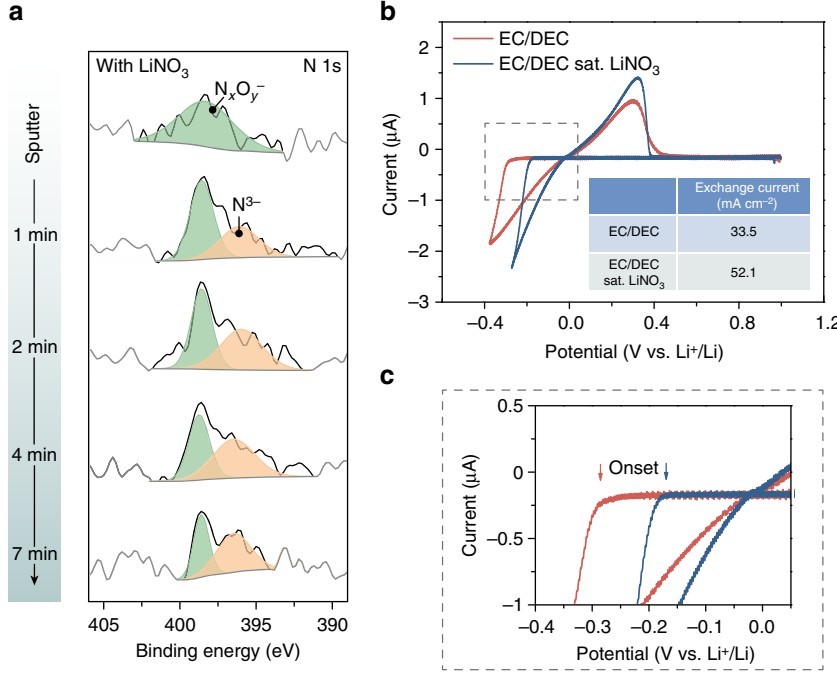

**Fig. 5** XPS and exchange current measurements. **a** XPS $N_{1s}$ depth profiles of the SEI formed in 0.5 M $LiPF_6$ EC/DEC electrolyte saturated with $NO_3^-$. Li deposition was carried out at a current density of 1 mA cm$^{-2}$ and a capacity of 0.1 mAh cm$^{-2}$ on Cu foil. **b** Typical CV scans (scan rate of 200 mV s$^{-1}$) obtained using ultramicroelectrode in 1.0 M $LiPF_6$ EC/DEC electrolyte with and without nitrate additive. **c** Enlarged CV scans around the nucleation potential region delineated in **b**. The presence of $NO_3^-$ in the electrolyte lowers the Li deposition onset potential by more than 100 mV

In conclusion, the physiochemical properties of the SEI layer, which largely rely on electrolyte composition, have a profound impact on the electrochemical cycling of metallic Li. In this study, we discuss the possibility of nitrate anions as an "electrolyte-agnostic" additive, which can be preferentially reduced during SEI formation to afford an interfacial environment conducive for controlled Li deposition, regardless of the electrolyte systems. It is discovered with surprise that $NO_3^-$ at millimolar-level in carbonate electrolytes can dramatically transform the morphology of Li nuclei from dendritic to spherical due to the significantly modified SEI properties. Structurally, a bilayered SEI can be observed with a dense inorganic shell, which effectively protects the Li metal surface and homogenizes the ion flux. Electrochemically, the nitrogen-containing species in the SEI result in improved ion-transport properties, and the presence of $NO_3^-$ in the Li-ion solvation sheath leads to faster charge transfer kinetics, both of which are favorable for containing dendrite growth. Moreover, a solubility-mediated sustained-release strategy is proposed to overcome the solubility limitation of nitrate salts in carbonates, and therefore enabling Li cycling efficiency and stability in corrosive carbonates on par with ether electrolytes. The distinct advantages of "electrolyte-agnostic" additive shed new light on electrolyte engineering principles without compromising the ionic conductivity, stability, and cost of the electrolyte, and can open up opportunities to the realization of high-energy-density batteries based on Li metal chemistries.

## Methods

**Fabrication of the LNO-SRF.** In all, 10 wt% PVDF-HFP (Kynar Flex 2801–00) and 5 wt% LiNO$_3$ (Alfa Aesar, approximately the maximum solubility of LiNO$_3$ in acetone) were fully dissolved in acetone by vigorous magnetic stirring on a 60 °C hotplate. The solution was then blade-coated on the rough side of electrodeposited Cu foil (CF-T8G-UN; Pred. Materials International, Inc.) using a 15-mil blade. Once the solvent was evaporated, the white-colored semi-transparent film can be peeled off from the Cu substrate. The obtained LNO-SRF was punched into 2 cm$^2$ round disks and transferred quickly into an argon-filled glovebox with sub-ppm O$_2$ and H$_2$O level (Vigor Tech) to minimize water absorption. The LNO-SRF was

dried on a 120 °C hotplate inside the glovebox for at least 1 week prior to use to prevent water contamination inside the battery.

**Electrochemical measurements.** 1.0 M LiPF$_6$ in EC/DEC electrolyte was purchased from BASF (Selectilyte LP40). 0.1 and 0.5 M LiPF$_6$ in EC/DEC electrolyte was made by diluting Selectilyte LP40 with EC (Acros Organics) and DEC (Acros Organics) solvents. LiTFSI (Solvay), LiFSI (Nippon Shokubai), DME (Sigma-Aldrich), FEC (Acros Organics), and VC (Acros Organics) were used as received. All the solvents were dried over 4A molecular sieves (Sigma-Aldrich) prior to making electrolytes.

Electrochemical testing was all carried out in 2032-type coin cell configuration. For studying the Li deposition morphology and cycling CE, the LNO-SRF was covered on top of the anode Cu current collector (25 μm, 99.8% metals basis; Alfa Aesar). Li foil (750 μm; Alfa Aesar) was used as the counter/reference electrode, two layers of Celgard 2325 (25 μm PP/PE/PP) was used as separators, and 80 μl electrolyte was added in each coin cell. The cells were rested for 3 h before testing to allow the LNO-SRF to fully swell in the electrolyte. For measuring the CE, the batteries were first subjected to 10 activation cycles between 0 and 1.5 V to form a stable SEI and eliminate side reactions before galvanostatic Li plating/stripping at different current densities and capacities on a LAND 8-channel battery tester. The LNO-SRF cells were rested 1 h between each plating/stripping cycle. For the Li|| NMC full-cell studies, either thin Li foil (~50 μm, Hydro-Québec) or Li-rGO was employed as the anode[8], and LNO-SRF was covered on the anode surface. The cathode was prepared by mixing LiNi$_{1/3}$Mn$_{1/3}$Co$_{1/3}$O$_2$ (NMC111, Xiamen Tungsten Co.), polyvinylidene fluoride (MTI), and carbon black (TIMCAL) at a ratio of 9:0.5:0.5 with $N$-methyl-2-pyrrolidone (Sigma-Aldrich) as the solvent. The cathode slurry was blade-coated on conductive carbon coated aluminum foil (MTI), calendared, and dried in 80 °C vacuum oven before use. All the Li||NMC batteries were tested between 2.7 and 4.3 V. When the voltage reached 4.3 V, a constant voltage charge process (4.3 V) was applied until the charge current decayed to 0.1 mA cm$^{-2}$. The EIS, CV, and exchange current density measurements were carried out on a Biologic VMP3 system. Detailed descriptions on the exchange current density measurements are provided in Supplementary Information.

**Characterization.** SEM images and EDX elemental mappings were taken with a FEI XL30 Sirion scanning electron microscope at an acceleration voltage of 5 and 20 kV, respectively. Before conducting SEM studies of Li electrodes, batteries were first disassembled in an argon-filled glovebox with sub-ppm O$_2$ and H$_2$O level, and then rinsed gently with DEC to remove residue Li salts. The samples were transferred into SEM with minimal air exposure. XRD patterns were recorded on a PANalytical X'Pert instrument. XPS analysis was obtained on a PHI VersaProbe 1 scanning XPS microprobe equipped with Al (Kα) source and argon ion sputter

gun with an air-free transfer vessel. The binding energies were calibrated with respect to the $C_{1s}$ peak at 284.5 eV. Argon sputtering for the XPS depth-profiling was carried out at a beam energy of 1 kV and current of 0.5 μA over a raster area of 2 mm × 2 mm. The sputtering rate calibrated on $SiO_2$ surface was ~2 nm min$^{-1}$.

Nitrate solubilities in different carbonate electrolytes were determined using a WestCo SmartChem 200 discrete analyzer. In a typical experiment, 200 μl electrolyte saturated with $LiNO_3$ was diluted to 5 ml with water. During analysis, nitrate anions in the solution were reduced quantitatively to nitrite by a cadmium column, diazotized with sulfanilamide, and subsequently coupled with N-(1-naphthyl)ethylenediamine dihydrochloride. The resulting colored azo dye was colorimetrically detected at 550 nm.

Cryo-EM characterizations were carried out on a FEI Titan 80–300 environmental (scanning) transmission electron microscope (E(S)TEM) operated at 300 kV following the cryo-transfer protocol described in our previous publication[29].

## Data availability

The datasets generated during and/or analyzed during the current study are available from the corresponding author on reasonable request.

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

## Acknowledgements

We acknowledge the support from the Assistant Secretary for Energy Efficiency and Renewable Energy, Office of Vehicle Technologies, Battery Materials Research (BMR) and Battery 500 Program of the U.S. Department of Energy. Yayuan L. acknowledges the support from Stanford University through the Stanford Graduate Fellowship. A.P. acknowledges support by the Department of Defense (DoD) through the National Defense Science & Engineering Graduate Fellowship (NDSEG) Program and support by the Stanford Graduate Fellowship.

## Author contributions

Yayuan L., D.L. and Y.C. conceived the concept and designed the experiments. Yayuan L., O.N. fabricated and characterized the materials with assistance from Yuzhang L. and Yanbin L. on cryo-EM, G.C. on discrete analyzer, and A.P. on ultramicroelectrode. Y.C. supervised the project. Yayuan L. and Y.C. prepared the manuscript with input from all the other coauthors.

## Additional information

**Competing interests:** The authors declare no competing interests.

