## [Peer Review File · Nature Communications]

Reviewers' comments:

Reviewer #1 (Remarks to the Author):

In the manuscript of NCOMMS-18-12440, the authors described the effect of nitrate anions on regulating Li deposition in carbonate-based electrolyte systems, which can alter the morphology of Li nuclei from dendritic to spherical. The authors attribute the effect to the preferential reductive decomposition of nitrate anions during SEI formation that modifies the interfacial environment. Additionally, a solubility-mediated sustained release methodology was introduced to encapsulate the nitrate anions to the PVDF-HFP porous polymer gel and the freestanding membrane can maintain a high concentration at the electroplating front. The performance of a battery has got significantly improved both in the half-cell and full-cell.

The understanding about the role of nitrate anions in carbonate-based electrolyte systems is novel, the experiments described here are well-conducted, and the manuscript is well written. Further suggestions and questions are provided as follow for improving the scientific rigor of the paper:

(1) In figure 1d, uniform spherical Li particles can be obtained in the presence of LiNO₃ at current density of 1.0 mA cm⁻² and a capacity of 0.1 mAh cm⁻². Whether the size of the spherical Li particles will become larger with the rise of capacity? If the size of the spherical Li particles reaches the limit, will they grow vertically or overlap to each other?

(2) While Li metal is applied in practical batteries, the requirement of the current density is 3.0 mA cm⁻² and 3.0 mAh cm⁻², so the morphology of the Li nucleation in the presence of LiNO₃ at current density of 3.0 mA cm⁻² is suggested, as well as the morphology of the Li deposition after 3.0 mAh cm⁻².

(3) The LNO-SRF was covered on top of the Cu current collector or Li surface to measure CE or full batteries. The thickness of the interlayer is ~18 μm, I also believe that much thinner film can also be prepared in practical batteries. How about the evolution of EIS in half cell or full cell during cycling?

(4) There are Li₂O and Li₂CO₃ characterized in Cryo-EM technique. The nitrogen species are also detected via EELS. A reasonable reaction equation is suggested here to describe the decomposed mechanism of LiNO₃.

(5) A stable SEI layer can be formed after initial cycle in carbonate-based electrolyte systems with LiNO₃. Is there any accumulating stress while Li plating? If so, whether the interphase will be broken by the accumulating stress.

(6) Electrochemical tests were performed either at 0.5 or 1.0 mA cm⁻² for both the Li-Cu and NCM cells in the manuscript. It is nice to provide a direct comparison about the capacity with increasing areal current density with or without LNO-SRF.

In summary, I'd like to recommend the manuscript to be published in Nature Communication after solving the issues above.

Reviewer #2 (Remarks to the Author):

In this manuscript, the authors demonstrated solubility-mediated sustained release of LiNO₃ in carbonate electrolytes from PVDF-HFP polymer matrix and thoroughly analyzed the resulting effect relating to progressive decomposition of LiNO₃. Theoretical explanations are provided as well, focusing on the mechanism for low solubility of LiNO₃ in carbonate solvents. The state-of-the-art characterization techniques, such as cryo-EM, for reactive lithium deposits are also systematically done and solidly prove the efficiency of the protection by LiNO₃. In addition, the beneficial effects of continuous release of LiNO₃ is evidenced with various control cell configurations. However, this work does not warrant the legitimate conceptual novelty compared to a recent paper (Proc. Natl. Acad. Sci. USA, 2018, 115, 5676). Although detailed procedure to engage progressive and sustainable release of

LiNO₃ is a bit different, the basic concept is very similar. Thus, I would feel that a more specialized journal is more appropriate.

Reviewer #3 (Remarks to the Author):

This manuscript reports the beneficial role of LiNO₃ in affecting the morphology of initially deposited Li and in improving the Coulombic efficiency and cycling stability for Li metal electrodes working in carbonate-based electrolyte. It has been shown that a LiNO₃/polymer composite film applied on the electrode surface can effectively address the solubility limitation of LiNO₃ and consequently render notable electrochemical performance of the Li metal anode. Despite a recently published article that reports a slightly similar utilization of controlled decomposition of LiNO₃ in carbonate electrolyte for stabilizing Li metal electrodes (Proc. Natl. Acad. Sci. U.S.A., 2018, 115, 5676-5680), this is concurrent work and possesses a significant amount of new knowledge complementary to the other study. In my opinion, this manuscript meets the standards of the journal and should be accepted for publication. The following comments and questions are for the authors to consider.

Considering the existence of at least some similarity, I recommend the authors to comment on the PNAS study in this manuscript.

The best electrode performance was achieved with the 18 micron thick LiNO₃/polymer composite film, whereas the TEM and XPS studies of the SEI layer were based on nitrate-saturated electrolyte. I am wondering whether and how, under different conditions as such, the SEI structure would differ, in terms of thickness, composition, chemical state, crystallinity, and so on. How might those differences, if they indeed exist, affect the electrode properties?

I would not be so sure that the N-containing species in the SEI are amorphous, because their content is so low. The atomic percentages can be derived from XPS or EELS. By the way, you may want to give the sputtering power and estimated sputtering rate for the XPS depth-profiling results.

Reviewer #1

In the manuscript of NCOMMS-18-12440, the authors described the effect of nitrate anions on regulating Li deposition in carbonate-based electrolyte systems, which can alter the morphology of Li nuclei from dendritic to spherical. The authors attribute the effect to the preferential reductive decomposition of nitrate anions during SEI formation that modifies the interfacial environment. Additionally, a solubility-mediated sustained release methodology was introduced to encapsulate the nitrate anions to the PVDF-HFP porous polymer gel and the freestanding membrane can maintain a high concentration at the electroplating front. The performance of a battery has got significantly improved both in the half-cell and full-cell.

The understanding about the role of nitrate anions in carbonate-based electrolyte systems is novel, the experiments described here are well-conducted, and the manuscript is well written. Further suggestions and questions are provided as follow for improving the scientific rigor of the paper. In summary, I'd like to recommend the manuscript to be published in Nature Communication after solving the issues above.

Response: We are excited that the reviewer agrees with us the importance of fundamentally understanding the role of nitrate anions on regulating the Li deposition morphology in carbonate-based electrolyte systems. We would like to thank the reviewer for the positive comments and the very constructive suggestions to further improve the quality of our manuscript.

(1) In figure 1d, uniform spherical Li particles can be obtained in the presence of LiNO_3 at current density of 1.0 mA cm^{-2} and a capacity of 0.1 mAh cm^{-2} . Whether the size of the spherical Li particles will become larger with the rise of capacity? If the size of the spherical Li particles reaches the limit, will they grow vertically or overlap to each other?

Response: We thank the reviewer for raising this important question. In general, galvanostatic deposition of Li metal will undergo two stages, namely, the nucleation stage and the growth stage. The nucleation stage (usually manifested as an initial overpotential bump) generates a fixed number of Li nuclei, beyond which the density of Li particles will not increase and instead, the particles will grow bigger as more Li is being deposited (Nano Lett., 2017, 17, 1132).

We carried out a detailed study in our work showing the morphological evolution of Li deposits with increasing capacities in the presence of LNO-SRF (**Figure 3a-c** of the manuscript). As can be observed from the SEM images, as the deposition capacity increased from 0.1 mAh cm^{-2} to 1 mAh cm^{-2} , the density of the Li particles remained the same. However, the size of the particles increased, and the morphology evolved gradually from densely packed spherical nuclei into nodule-like shapes. Such observation is consistent with the classical nucleation and growth theory.

(2) While Li metal is applied in practical batteries, the requirement of the current density is 3.0 mA cm^{-2} and 3.0 mAh cm^{-2} , so the morphology of the Li nucleation in in the presence of LiNO_3

at current density of 3.0 mA cm^{-2} is suggested, as well as the morphology of the Li deposition after 3.0 mAh cm^{-2} .

Response: We appreciate the reviewer's valuable suggestion on checking the Li deposition morphology under conditions employed in practical batteries. The SEM images of Li deposition in the presence of LNO-SRF at a current density of 3 mA cm^{-2} are shown in Figure R1.

Figure R1. Li deposition morphology at a current density of 3 mA cm^{-2} and a capacity of (a) 0.1 mAh cm^{-2} (nucleation), and (b) 3 mAh cm^{-2} .

At higher deposition current density, the Li nuclei in the presence of nitrate remained spherical. However, the nucleation density was increased (the size of the Li nuclei was much smaller). This is consistent with the Li nucleation theory under galvanostatic condition (Nano Lett., 2017, 17, 1132), where higher current density results in higher deposition overpotential and the nuclei size is inversely proportional to the overpotential. Instead of dendritic Li, dense Li particles can still be observed after 3 mAh cm^{-2} deposition in the presence of nitrate. This can further account for the significantly improved Li cycling efficiency at practical current densities and capacities observed in our study (Supplementary Fig. 14).

(3) The LNO-SRF was covered on top of the Cu current collector or Li surface to measure CE or full batteries. The thickness of the interlayer is $\sim 18 \text{ }\mu\text{m}$, I also believe that much thinner film can also be prepared in practical batteries. How about the evolution of EIS in half cell or full cell during cycling?

Response: It is a great idea to evaluate the stability of the Li metal interphase during long-term cycling via EIS. Correspondingly, we carried out EIS measurements on Li||Li symmetric cells cycled at a current density of 1 mA cm^{-2} and a capacity of 1 mAh cm^{-2} with or without LNO-SRF. As can be observed from Figure R2, without the presence of nitrate in the electrolyte, the charge transfer impedance of the symmetric cell after 50 cycles decreased dramatically due to the excessive formation of Li dendrites that roughened the electrode surface. On the other hand, with the sustained concentration of nitrate in the electrolyte, the impedance of the symmetric cell remained much more stable after long-term cycling, confirming the benefits of nitrate anions on stabilizing the Li metal surface and suppressing dendrite formation.

Figure R2. Electrochemical impedance spectra of Li||Li symmetric cells cycled at a current density of 1 mA cm^{-2} and a capacity of 1 mAh cm^{-2} with and without the presence of LNO-SRF.

(4) There are Li_2O and Li_2CO_3 characterized in Cryo-EM technique. The nitrogen species are also detected via EELS. A reasonable reaction equation is suggested here to describe the decomposed mechanism of LiNO_3 .

Response: We agree with the reviewer that it is important to describe the decomposition mechanism of LiNO_3 . As discussed in the manuscript, the dense, crystalline Li_2O outer layer of the SEI observed under cryo-EM is attributed to the fast decomposition of LiNO_3 on Li metal surface. The following reactions are possible for LiNO_3 decomposition:

The initial SEI layer formed by nitrate decomposition cannot completely prevent the permeation of electrolyte given the volume change of Li metal during deposition, such that additional reactions between the solvents and Li result in the amorphous inner layer, until the electron tunneling limit is reached. Therefore, the Li_2CO_3 component comes from the decomposition of the carbonate electrolyte.

As discussed in the classical review paper of professor Kang Xu (Chem. Rev., 2004, 104, 4303), electrolyte decomposition is an extremely complicated process with multiple possible pathways, some common decomposition reactions of ethylene carbonate (EC) are listed below, which generate Li_2CO_3 as a product:

(5) A stable SEI layer can be formed after initial cycle in carbonate-based electrolyte systems with LiNO_3 . Is there any accumulating stress while Li plating? If so, whether the interphase will be broken by the accumulating stress.

Response: We thank the reviewer for raising this important point. We believe that new SEI forms in each Li plating/stripping cycle, and there is little driving force for Li metal to be plated back into the old SEI shell. The formation of new SEI in each plating/stripping cycle is, therefore, a major source of Li cycling inefficiency. The role of electrolyte additive is to improve the quality of the SEI, such that denser/thinner SEI can be formed, which suppresses the continuous corrosion of Li metal by the electrolyte to improve the cycling efficiency.

To reduce the repeated breakdown/reformation of the SEI, other Li metal surface protection techniques can be employed in combination with electrolyte additives, such as artificial SEI coatings. These coating layers can prevent the direct contact between liquid electrolyte and Li metal surface, as well as provide mechanical support to the native SEI, which is beneficial to enhance the interphase stability.

(6) Electrochemical tests were performed either at 0.5 or 1.0 mA cm^{-2} for both the Li-Cu and NCM cells in the manuscript. It is nice to provide a direct comparison about the capacity with increasing areal current density with or without LNO-SRF.

Response: We appreciate this constructive suggestion from the reviewer. In fact, in our original manuscript, electrochemical tests on Li-Cu half cells were carried out at a variety of current densities and capacities (up to 2 mA cm^{-2} and 3 mAh cm^{-2} , Supplementary Fig. 14). Electrochemical tests on Li-NMC full cells were carried out using both low capacity (Fig. 3g) and high capacity (Supplementary Fig. 20) cathodes at current densities of 0.2 mA cm^{-2} , 0.5 mA cm^{-2} , 1 mA cm^{-2} , and 1.5 mA cm^{-2} . In addition to these, we also carried out additional electrochemical cycling using high mass loading NMC at a high current density of 3 mA cm^{-2} . As can be observed from Figure R3, with the sustained release of nitrate anions (LNO-SRF film) and the employment of three-dimensional Li (Li-rGO) to dissipate the local current density on Li metal surface, a much higher cathode capacity as well as improved cycling stability can be realized.

Figure R3. Cycling performance of NMC||Li full cell with high mass loading cathode ($\sim 20 \text{ mg cm}^{-2}$) at a current density of 3 mA cm^{-2} . The red data points correspond to NMC cycled with a large amount of excess Li (750 μm Li foil) in 0.5 M LiPF_6 EC/DEC electrolyte with 2% VC additive. The blue data points

correspond to NMC cycled with limited excess Li (150 μm Li-rGO composite anode) in the presence of LNO-SRF in 0.5 M LiPF_6 EC/DEC electrolyte.

Reviewer #2

In this manuscript, the authors demonstrated solubility-mediated sustained release of LiNO_3 in carbonate electrolytes from PVDF-HFP polymer matrix and thoroughly analyzed the resulting effect relating to progressive decomposition of LiNO_3 . Theoretical explanations are provided as well, focusing on the mechanism for low solubility of LiNO_3 in carbonate solvents. The state-of-the-art characterization techniques, such as cryo-EM, for reactive lithium deposits are also systematically done and solidly prove the efficiency of the protection by LiNO_3 . In addition, the beneficial effects of continuous release of LiNO_3 is evidenced with various control cell configurations. However, this work does not warrant the legitimate conceptual novelty compared to a recent paper (Proc. Natl. Acad. Sci. USA, 2018, 115, 5676). Although detailed procedure to engage progressive and sustainable release of LiNO_3 is a bit different, the basic concept is very similar. Thus, I would feel that a more specialized journal is more appropriate.

Response: Firstly, we would like to thank the reviewer for acknowledging the thoroughness of our study. Indeed, our work not only utilized a combination of state-of-the-art characterization techniques to understand the effect of LiNO_3 decomposition on the physiochemical properties of SEI and the corresponding metallic Li plating behavior, but also proposed a continuous release strategy to overcome the solubility barrier of LiNO_3 in carbonate electrolytes and therefore, significantly improved the electrochemical performance of Li metal half and full cells.

We also understand the reviewer's concern regarding the other work published very recently, and we sincerely hope that we could address such concern from the following perspectives:

- (1) Our work and the other work published recently (Proc. Natl. Acad. Sci. USA, 2018, 115, 5676; hereinafter referred to as "the PNAS work") are **concurrent works developed independently by the two labs**. As the reviewer might also noticed, our work was submitted on Apr. 24th to Nature Communications, while the PNAS work was published online on May 14th. Therefore, we were unaware of the PNAS work neither when we carried out the study nor at the time we submitted the manuscript. Nevertheless, we are glad that the researchers of the PNAS work also observed appreciably improved electrochemical performance of Li metal anode in carbonate electrolytes the presence of LiNO_3 , which further corroborated our claims that the preferential reduction of nitrates during SEI formation can afford an interfacial environment conducive for controlled Li deposition (a uniform, bilayered SEI structure with fast ion transport and high exchange current density). It will be truly unfortunate if there shall be a twist of fate due to slight timing offset and we hope works developed independently around the same time in the scientific community can be treated equally.
- (2) Despite of similarity to some extent between our work and the PNAS work on utilizing the concept of sustained release of LiNO_3 in carbonates for stabilizing Li metal anode, we are confident that **our work brings considerably more insights into the fundamental**

understandings of the additive effects, as also acknowledged by Reviewer #3. Though impressive electrochemical performance was observed in the PNAS work (100 cycle average Coulombic efficiency of 96.8% at a current density of 2 mA cm^{-2} and a high capacity of 5 mAh cm^{-2}), little efforts were made to explain the improved performance, with the effect being simply attributed to the possible formation of “ Li_3N and LiN_xO_y that are known to be good Li ion conductors and can promote efficient and stable cycling of Li metal electrodes”.

On the other hand, **our work is devoted to elucidating the intrinsic mechanism of nitrate on modifying the SEI properties**, as one can tell from the extensive discussion section of the manuscript. Experimental and theoretical evaluations were provided to explain the origin of the limited nitrate solubility in carbonates. Moreover, with the help of cryogenic transmission electron microscopy (cryo-EM), x-ray photoelectron spectroscopy (XPS), ultramicroelectrode and electrochemical impedance spectroscopy (EIS), we were able to develop a solid explanation on how the preferential reduction of nitrates can create a desirable interfacial environment for controlled Li metal deposition. Namely, the presence of nitrate resulted in a pronounced transformation of SEI from amorphous to a bilayered structure, as revealed by cryo-EM. The outer layer consisted of highly ordered crystalline Li_2O due to nitrate decomposition, which can not only improve the homogeneity of the SEI to promote uniform ion flux, but also effectively shield the Li surface from electrolyte corrosion. XPS showed the strong presence of LiN_xO_y and Li_3N in the inner layer of the SEI, which facilitates fast ion transport, as also confirmed with the EIS results. The ultramicroelectrode revealed doubled exchange current of the Li^+/Li couple with nitrate additive, due to the presence of nitrate anions in the Li^+ solvation sheath. This resulted in increased nucleation exclusion zone, and thus bigger Li particles with reduced surface area.

We are convinced that a top journal like *Nature Communications* shall not be confined to merely observing improved performance by material design. It is undoubtedly more meaningful if we could **understand the fundamentals of the SEI, which remains largely elusive to the battery community.** It is certain that the new understandings here will bring significant new insight to the community and guide the future development of advanced electrolytes (as mentioned in our manuscript, the beneficial effects were also observed with nitrites, and could possibly to extended to nitrogen-containing organic compounds or sulfides).

- (3) In addition to the two main arguments mentioned above, our work also shows distinct advantages compare to that reported the PNAS work in terms of the **material design** for the continuous release of LiNO_3 . Some major points are:
- a) In the PNAS work, the authors immersed glass fiber separator in a LiNO_3 solution to impregnate the separator with LiNO_3 crystallites with sizes of a few hundred nanometers. In our work, by dispersing LiNO_3 in a polymeric matrix, much smaller particles ($\sim 60 \text{ nm}$, Supplementary Fig.8) and more uniform distributions were realized, which can

significantly enhance the dissolution kinetics in electrolytes, as proved in Supplementary Fig.7.

- b) Battery is a delicate system that requires the optimization of each components. Another highlight of our work is to explore the influence of electrolyte molarity on the effectiveness of nitrate additive. For example, reducing the LiPF_6 concentration results in higher NO_3^- solubility, such that nitrate additive in 0.5 M LiPF_6 ethylene carbonate/diethyl carbonate electrolyte resulted in much higher Li metal cycling efficiency compare to the 1 M counterpart, while further decreasing the LiPF_6 concentration deteriorated the battery performance (Supplementary Fig.12 and 13).
 - c) Instead of employing a relatively thick glass fiber separator (260 μm , reactive to metallic Li), a thin polymeric coating ($\sim 18 \mu\text{m}$) that can be applied directly on commercial battery separators shows a much better compatibility with the state-of-the-art battery assembly process.
 - d) We demonstrated the feasibility of pairing our material with $\text{LiNi}_{1/3}\text{Mn}_{1/3}\text{Co}_{1/3}\text{O}_2$ (NMC) cathode, a highly promising next-generation cathode material, which is more relevant to the battery community than the Li-MoS_3 chemistry demonstrated in the PNAS work.
- (4) In an effort to acknowledge the PNAS work, have added the following discussion on it in the introduction section of the manuscript (highlighted as below).

“Note that during the peer-review process of our manuscript, we noticed another work utilizing a slightly similar engineering method to overcome the solubility barrier of LiNO_3 in carbonate electrolyte, which also resulted in improved Li anode CE. Nevertheless, we are confident that our comprehensive work brings a significant amount of new knowledge complementary to the other study. Particularly, the mechanism of NO_3^- in modifying the SEI properties was studied deliberately on the basis of cryo-electron microscopy (cryo-EM), ultramicroelectrode, x-ray photoelectron spectroscopy (XPS), and electrochemical impedance spectroscopy (EIS). The characterizations unraveled...”

Reviewer #3

This manuscript reports the beneficial role of LiNO_3 in affecting the morphology of initially deposited Li and in improving the Coulombic efficiency and cycling stability for Li metal electrodes working in carbonate-based electrolyte. It has been shown that a LiNO_3 /polymer composite film applied on the electrode surface can effectively address the solubility limitation of LiNO_3 and consequently render notable electrochemical performance of the Li metal anode. Despite a recently published article that reports a slightly similar utilization of controlled decomposition of LiNO_3 in carbonate electrolyte for stabilizing Li metal electrodes (Proc. Natl. Acad. Sci. U.S.A., 2018, 115, 5676-5680), this is concurrent work and possesses a significant amount of new knowledge complementary to the other study. In my opinion, this manuscript meets the standards of the journal and should be accepted for publication. The following comments and questions are for the authors to consider.

Response: We are beyond grateful to the reviewer for the positive comments on our manuscript, and importantly, for acknowledging the new insights our work brought complementary to the other concurrent work (Proc. Natl. Acad. Sci. U.S.A., 2018, 115, 5676; hereinafter referred to as “the PNAS work”) published very recently.

Indeed, our work and the PNAS work are works developed independently by the two labs at around the same time. As the reviewer might also noticed, our work was submitted on Apr. 24th to Nature Communications, while the PNAS work was published online on May 14th. Therefore, we were unaware of the PNAS work neither when we carried out the study nor at the time we submitted the manuscript. Nevertheless, we are glad that the researchers of the PNAS work also observed appreciably improved electrochemical performance of Li metal anode in carbonate electrolytes the presence of LiNO_3 , which further corroborated our claims that the preferential reduction of nitrates during SEI formation can afford an interfacial environment conducive for controlled Li deposition (a uniform, bilayered SEI structure with fast ion transport and high exchange current density). Despite utilizing slightly similar engineering method to overcome the solubility barrier of nitrate, we are confident that our work can bring considerably more insights into the fundamental understandings of the additive effects, which the PNAS work shed little light on. Therefore, we would sincerely thank the reviewer for treating our concurrent work equally and objectively.

(1) Considering the existence of at least some similarity, I recommend the authors to comment on the PNAS study in this manuscript.

Response: We appreciate the valuable suggestion from the reviewer. Correspondingly, we have added the following discussion on the PNAS work in the introduction section of the manuscript (highlighted as below).

“Note that during the peer-review process of our manuscript, we noticed another work utilizing a slightly similar engineering method to overcome the solubility barrier of LiNO₃ in carbonate electrolyte, which also resulted in improved Li anode CE. Nevertheless, we are confident that our comprehensive work brings a significant amount of new knowledge complementary to the other study. Particularly, the mechanism of NO₃⁻ in modifying the SEI properties was studied deliberately on the basis of cryo-electron microscopy (cryo-EM), ultramicroelectrode, x-ray photoelectron spectroscopy (XPS), and electrochemical impedance spectroscopy (EIS). The characterizations unraveled...”

(2) The best electrode performance was achieved with the 18-micron thick LiNO₃/polymer composite film, whereas the TEM and XPS studies of the SEI layer were based on nitrate-saturated electrolyte. I am wondering whether and how, under different conditions as such, the SEI structure would differ, in terms of thickness, composition, chemical state, crystallinity, and so on. How might those differences, if they indeed exist, affect the electrode properties?

Response: We thank the reviewer for raising this very interesting question.

All the characterizations on the physiochemical and electrochemical properties of the SEI were carried out using nitrate-saturated electrolyte to avoid the contamination of the Li metal surface by the polymeric species from the gel electrolyte film. We took special care to make sure that the TEM and XPS were done on samples at extremely small Li deposition capacity (0.02 mAh cm⁻² for TEM and 0.1 mAh cm⁻² for XPS), where the morphology of the Li deposits (nuclei) remained spherical. This is a regime that we believe the concentration of nitrate in the electrolyte is not a limiting factor and is sufficient enough to dominate the SEI formation process. Once the nucleation stage is passed, the Li particles will grow bigger with increasing deposition capacity, together with the continuous breakdown and reformation of new SEI. Without the LiNO₃/polymer composite film, the nitrate anion will be depleted in the electrolyte and therefore, unable to further dominate the SEI formation process, which account for the observation of Li dendrites at higher deposition capacities in nitrate-saturated electrolyte. On the other hand, with the LiNO₃/polymer composite film, nitrate concentration at the electroplating front can remain high to continuously form desirable SEI structure for controlled Li deposition.

Therefore, from the above discussion, we do not think the different conditions (saturated electrolyte vs. LiNO₃/polymer composite film) would induce a significant difference in the SEI.

(3) I would not be so sure that the N-containing species in the SEI are amorphous, because their content is so low. The atomic percentages can be derived from XPS or EELS. By the way, you

may want to give the sputtering power and estimated sputtering rate for the XPS depth-profiling results.

Response: We agree with the reviewer that we might not have sufficient evidence to draw a firm conclusion that the N-containing species in the SEI are amorphous. We reached the conclusion from the fact that neither crystalline lattices nor diffraction patterns belonging to N-containing species can be observed from our TEM studies. But it is also possible that the concentration of these species is too low to afford detectable domains under TEM. For the sake of scientific rigor, we added the following to the manuscript:

“...Thus, the nitrogen species from the preferential reduction of NO_3^- shall remain amorphous in the SEI (or, the size of their domains are too small to be observed by TEM possibly due to limited concentration)...”

Moreover, we thank the reviewer for the constructive comment on including the XPS sputtering power and rate in the manuscript. Argon sputtering for the XPS depth-profiling was carried out at a beam energy of 1 kV and current of 0.5 μA over a raster area of 2 mm * 2 mm. Due to the extreme air sensitivity of both Li metal and SEI, it is challenging to carry out sputtering rate calibration on them directly. The sputtering rate calibrated on SiO_2 surface was ~ 2 nm/min.

We have also included the above information in the Method section of the manuscript.

REVIEWERS' COMMENTS:

Reviewer #1 (Remarks to the Author):

The authors have revised the manuscript very carefully. It can be accepted now.

Reviewer #3 (Remarks to the Author):

The revisions have addressed my comments. I recommend publishing the manuscript as is.